# Ultrastructural Changes of Blood Cells in Children with Generalized Purulent Peritonitis: A Cross-Sectional and Prospective Study

**DOI:** 10.3390/children7100189

**Published:** 2020-10-17

**Authors:** Ulyana Halyuk, Olena Lychkovska, Oksana Mota, Vasyl Kovalyshyn, Natalia Kech, Petro Pokotylo, Olena Trutiak, Bożena Zboina, Grzegorz Józef Nowicki, Barbara Ślusarska

**Affiliations:** 1Department of Normal Anatomy, Lviv National Medical University, UA-79010 Lviv, Ukraine; uhalyuk@gmail.com (U.H.); mota.oksana@gmail.com (O.M.); VasylKovalyshyn1950@gmail.com (V.K.); anatompetro@gmail.com (P.P.); 2Department of Propaedeutic Pediatrics and Medical Genetics, Lviv National Medical University, UA-79010 Lviv, Ukraine; Olychkovska@gmail.com (O.L.); olenatrutiak@gmail.com (O.T.); 3Institute of Hereditary Pathology, National Academy of Medical Sciences of Ukraine, UA-79000 Lviv, Ukraine; NataliaKech@ukr.net; 4Department of Pedagogy and Health Sciences, College of Business and Entreprise, PL-27-400 Ostrowiec Świetokrzyski, Poland; bozenazboina@poczta.fm; 5Department of Family Medicine and Community Nursing, Medical University of Lublin, PL-20-081 Lublin, Poland; gnowicki84@gmail.com

**Keywords:** purulent peritonitis, acute appendicitis, blood cells, ultrastructural changes

## Abstract

In conditions of abdominal sepsis with indications of first- or second-stage shock, blood cells undergo significant ultrastructural changes that cause impaired gas exchange, changes in reactivity, and decompensation of organs and systems functions. This paper presents a cross-sectional prospective study aimed at researching the ultrastructure of blood cells in children experiencing abdominal septic shock against the background of generalized purulent peritonitis of appendicular origin. This study was conducted with 15 children aged 6–12 who were undergoing treatment for generalized appendicular purulent peritonitis, with first- or second-stage abdominal septic shock, in emergency care. The changes in the ultrastructure of erythrocytes did not correspond to changes characteristic of eryptosis, which confirms their occurrence under the influence of such pathogenic factors as intoxication, metabolic, water–electrolyte balance, and acid–base disorders. Ultrastructural changes of granulocytes indicate their hyperactivation, which leads to the exhaustion of membrane synthetic resources, membrane destruction, ineffective expenditure of bactericidal factors on substrates that are not subject to destruction. In lymphocytes, disorganization of the nuclear membrane and intracellular membranes, uneven distribution of chromatin, the hypertrophied Golgi apparatus, and a large number of young mitochondria, lysosomes, ribosomes, vesicles manifesting the disruption of metabolism, stress and decompensation of energy supply and protein synthesis systems, have been demonstrated. In conditions of abdominal sepsis with indications of first- or second-stage shock, blood cells undergo substantial ultrastructural changes causing gas exchange disruption, changes in reactivity, as well as decompensation of organs and system functioning.

## 1. Introduction

Acute appendicitis (AA) is the most common surgical emergency the world over [1], yet it is difficult to diagnose, especially in women, for whom there is a broader range for differential diagnosis [2]. Every year this illness appears in 90–100 out of 100,000 patients living in developed countries. The risk for developing AA during one’s lifetime is 8.6% for men and 6.7% for women, with the illness most often occurring between the ages of 10 and 30 [3]. In a study by Omling et al. [4] conducted on a group of 38,939 children, the estimated risk of the appearance of AA in patients under age 18 was 2.5%. In epidemiological studies, geographical variations have been noted in the incidence of the illness over the course of a lifetime, with a frequency of 16% in South Korea, 9% in the USA, and 1.8% in Africa [5,6].

The immediate cause of inflammation of the appendix is obstruction of the appendiceal lumen. Furthermore, there are several infectious agents that may cause this illness or be associated with it [7,8]. The entire scope of causative factors of acute appendicitis, however, is not known [9]. The most recent theories on the cause of AA concentrate on genetic factors [10], influence of environment [11], and infections [12]. Diagnosing AA requires conducting a medical interview with the patient or his/her caregivers, a physical exam and laboratory tests, supplemented by diagnostic imaging. There is no consensus on the optimal diagnostic path in the case of acute right iliac fossa (RIF) pain, as to diagnostic imaging: whether ultrasonography (USG), computed tomography(CT), or magnetic resonance imaging(MRI) [13]. Exposure to radiation, however, excludes some of them from routine use [14]. For several years, scientific societies have tried to develop clinical guidelines, confirmed by scientific evidence, meant to standardize the diagnostic and therapeutic procedure for AA, but not all provisions have been universally accepted. According to a survey conducted by Zani et al. [15], in a group of 169 delegates from 42 countries, including 24 European nations, at the annual congress of the European Paediatric Surgeons’ Association (EUPSA), in diagnosing AA the majority of the surgeons surveyed relied on complete blood count (CBC) findings (92%), C-reactive protein (82%) and ultrasound of the abdominal cavity (76%). CT or MRI scans, on the other hand, were indicated less frequently.

AA often causes acute abdominal pain and can lead to perforation and peritonitis. Generalized purulent peritonitis (GPP) caused by destructive forms of appendicitis is one of the burning issues associated with emergency pediatric surgery and intensive therapy. Unfavorable conditions of the pathology may lead to development of abdominal sepsis, which still has a high mortality rate despite encouraging progress in emergency antibiotic therapy and the adoption of new surgical methods for this type of patient [16,17]. Sepsis is a life-threatening organ dysfunction caused by dysregulated host response to infection [18]. Research into molecular causes of sepsis revealed significantly more differentiated and complex dependencies between an infectious factor and the organism, which collectively cause heterogenic symptoms of septicemia. Hemodynamic symptoms of sepsis lead to development of septic shock. According to ESICM 2014, shock is a generic term for acute failure of bloodstream which poses a threat to life and is linked to insufficient oxygen consumption by cells. This is the condition when the cardiovascular system does not provide tissues with oxygen adequate to satisfy their needs (although respiratory insufficiency or anemia is not the only or domineering reason for reduced oxygen transportation). As a result, there is dysoxia at the cellular level (i.e., the independence between the utilization of oxygen and its supply is lost), causing an increase in both anaerobic metabolism and lactate production [19].All changes developed in shock at the molecular, cellular and clinical levels connected with hypoxemia, hypovolemia/hypotension associated with immune response, metabolism, hemodynamics, coagulopathy, and/or neuroendocrine response disrupt tissue perfusion, oxygen transport and ATP synthesis, which leads to diminished availability of energy, nutrients and finally hypoxia. This causes serious damage to cells [20], which results in structural changes of peripheral blood. Clinical and experimental studies indicate that sepsis causes a range of effects for erythrocytes. They are as follows: sepsis changes heterogeneity of red blood cell distribution width (RDW) [21], can move acurve of haemoglobin oxygen dissociation to the left [22], and causes changes of erythrocytes morphology, in particular rheology—deformability of red blood cells (RBC), which can be an early indicator of septicemia [23]. Sepsis also causes a serious disorder of the immune response to an infection that results in neutrophil dysfunction [24]. Research into clinically significant biomarkers of septicemia have been commenced on and it has not yielded effective results yet; thus, there is need for investigations into this area. Despite emerging research on the ultrastructure of blood cells for adults in critical condition [25], there is lack of similar studies of children. The purpose of this study is to research the ultrastructure of blood cells in children in abdominal septic shock against the background of generalized purulent peritonitis of appendicular origin.

## 2. Materials and Methods

### 2.1. Study Desing and Population

A cross-sectional and prospective study was conducted from November 2018 to March 2019 on 15 children being treated for appendicitis-related generalized purulent peritonitis at the Emergency Department of the Lviv Regional Children’s Hospital Okhmatdyt. The inclusion criteria were: abdominal sepsis with signs and symptoms of first- or second-stage shock, emergency surgery, histopathological confirmation of acute appendicitis and generalized peritonitis, and written informed consent of the child’s parents/guardians for participation in the study. Prior to the qualification for the study participation, all the patients were assessed with the following tools: the SIRS (Systemic Inflammatory Response Syndrome) [26] and the SOFA (Sequential Organ Failure Assessment Score) [18]. The results obtained based on the two scales are presented in the Appendix A. Exclusion criteria were: generalized peritonitis of a different cause (e.g., complicated diverticulitis), or lack of written informed consent for participation in the study.

The research project received approval from the Bioethics Commission of the Medical University of Lublin (KE-0254/197/2017) and was conducted in accord with the Declaration of Helsinki. All guardians (parents, legal guardians) of the respondents were presented with the purpose of the study, and then asked for their written consent to participate in it.

### 2.2. Participants

From November 2018 to March 2019, a total of 39 children (aged up to 18 years old) were admitted to the Emergency Department of the Lviv Regional Children’s Hospital Okhmatdyt. The pediatric patients were qualified for emergency surgery due to appendicitis based on the Alvarado score of ≥7 points [27]. Owing to the histopathologic examination performed after the surgery, 24 patients were excluded from the study because of the lack of diagnosis of generalized purulent peritonitis. Finally, the study encompassed 15 children who needed surgery because of generalized purulent peritonitis of appendicular origin. The flow chart defining the final group included in the study is presented in Figure 1.

### 2.3. Blood Sample

Blood samples were taken from the elbow veinprior to the surgery. The blood was collected into two sterile vacuum EDTA-K3 collection tubes and one tube with clotting activator and separating agent (granules). The collection was performed by a trained nurse. The tubes were provided to the laboratory within 15 min after the collection and directly after the provision to the lab, the examination was begun. Until the tubes were transported to the lab, they were stored in the fridge at temperature from +2 to +8 °C and protected from light. The material from one test tube was used for a complete blood count. The blood tests were conducted using an automatic hemolytic analyzer HTI MicroCC-20 Plus (Doc. Num. OM-R-MCC-20-H). Centrifuged serum from the second tube was used to determine the total bilirubin and C-reactive protein using standard laboratory methods.

Material from the second test tube: the blood sampling was conducted on a standard solution of glugicyr with the addition of glutaraldehyde and paraformaldehyde. After centrifugation, the precipitate was fixed at 2% OsO4 solution on the monophosphate buffer. After repeated centrifugation and washing of the precipitate, it was dehydrated in ethanol and acetone in increasing concentrations. Next, the precipitate was poured into eponaraldite and the sections were contrasted in 2% uranyl acetate solution followed by lead citrate solution. The examining and photographing of the material was done with a УEMB-100K (Sumy, Ukraine) microscope at an accelerating voltage of 75 kV and screen magnification of 2000×–12400×.

### 2.4. Other Data

Other data—age, gender, and duration of symptoms—were collected by means of standard questionnaires used in personal interviews. The interviews were made with the patient’s parent or legal guardianin the preoperative and perioperative period.

### 2.5. Statistical Analysis

The statistical analysis included the calculation of mean and standard deviations (SD) or median (interquartile range, IQR) as appropriate. For non-measurable parameters, the estimated frequencies and percentages were used. For measurable characteristics, the normality of the distribution was checked with the Shapiro–Wilk test. The analyses were conducted using IBM SPSS Statistics for Windows, version 25 (IBM Corp., Armonk, NY, USA).

## 3. Results

### 3.1. Characteristics of Participants

Table 1 presents the characteristics of the study group. A total of 15 children from 6 to 12 years old being treated for purulent generalized peritonitis took part in the study. There were nine boys in the group, and the average duration of symptoms was 50.67 (SD = 18.6) hours.

### 3.2. Results of the Electron Microscopy Analysis of Blood Sediment of the Examined Children in a State of Sepsis

The electron-microscopic analysis of blood precipitate under septic state has revealed that it mainly contains erythrocytes and granulocytes with much smaller number of lymphocytes. Figure 2 and Figure 3 present sample images from microscopic electron analysis of selected elements of blood sediment from the patients being studied.

The erythrocytic mass was represented chiefly by cells of irregular shape (Figure 2a–d). According to Moroz et al. [25], they are chiefly stomatocytes and echinocytes with few erythrocytes in their common morphology of biconcave discs. According to O’Connor [28], they are chiefly stomatocytes and echinocytes belong to degenerative forms. As for the echinocytes (burr cells), they are presented by cells with numerous thorny projections. The hyaloplasm of the detected echinocytes displayed low electronic density. The surrounding plasmalemma was either significantly loosened or almost lacking, thus disorganizing the peripheral layers of the cytoplasm, the individual elements of which were dissected from the mass (Figure 1c,d). Furthermore, all the erythrocytes of irregular shape had areas where plasmalemma was not only disorganized but even completely absent (Figure 2c). Judging from their electron density, the erythrocytes were heteromorphic. Smaller erythrocytes with separate digitiform projections had the highest electron density (Figure 2b,c). Such changes are characteristic of apoptosis. However, most of the erythrocytes produced changes, such as a decrease in cell volume and its condensation (pyknosis), which do not correspond to those described for eryptosis. By contrast, most of the erythrocytes had low electron density and exhibited signs of cytoplasm disruption accompanied by the formation of numerous lumps of electron compact material.

Dramatic changes also occurred in cells of the granulocytic series in all examined children experiencing generalized purulent peritonitis. Considerable degranulation was accompanied by disorganization of intracellular membranes, even sometimes by decomposition of cytoplasm (Figure 3a). The most drastic changes were produced in neutrophils (Figure 3a–c). Their nuclear membranes were either mostly disorganized or absent. In places of their complete absence, flecks of chromatin were spreading out into the cytoplasm. Cytoplasm was presented as mass of electron-light hyaloplasm, which began disintegrating as soon as drops of fat appeared. Cytoplasmic granules are small in size and contain a small amount of electron-dense material. In the neutrophilic series, there are also granules lacking their content. Not far from the hyaloplasm areas, microvesicles showing signs of disintegration were discovered. Cells with the above-described ultrastructure correspond to level two neutrophil activity or hyperactivity [29].

Lymphocytes of all the examined children were notable for their uneven distribution of chromatin, which is mostly represented by heterochromatin (Figure 3d). Nuclear membranes were mostly missing. There was a hypertrophied Golgi apparatus, as well as a great number of young mitochondria, lysosomes, ribosomes, and vesicles. Plasmalemma, like all the rest of intracellular membranes, exhibited signs of disorganization. Such ultrastructural changes correlate with lymphocytes actively participating in immune response.

## 4. Discussion

The results of our study in children suffering from abdominal sepsis with first- and second- stage shock show the changes of ultrastructure of erythrocytes, granulocytes, and lymphocytes. The specific changes of erythrocytes included: irregular shape of the echinocytes (burr cells), diminished volume of cells, condensation (pyknosis), low density of electrons and signs of rupture of cytoplasm. The specific changes of granulocytes ultrastructure encompassed significant degranulation with accompanying disorganization of intracellular membranes and sometimes even with decomposition of cytoplasm. Lymphocytes of the children examined were characterized by uneven distribution of chromatine and lack of nuclear membranes in the cells. Apart from the aforementioned changes, the Golgi apparatus was larger and there was a high number of young mitochondria, lysosomes, ribosomes and vesicles. Under normal physiological conditions, the mature erythrocyte has a characteristic biconcave disc shape, approximately 6–8 µm in diameter and 2 µm thick [30]. Sepsis alters morphology of erythrocytes and affects aggregation of RBC. Presence of echinocytes (RBC)with numerous spikes protruding from the surface) was found in the study by De Oliveiret al. [31], aggregates of RBC and improper RBC in a patient with sepsis in the study by Piagnerelliet al. [32]. Sepsis can cause an increase or decrease in WBCs and triggers the release of cytokines, which results in the release of immature leukocytes from bone marrow. The structural changes of granulocytes and lymphocytes are rarely described in literature; thus, our results can serve as the basis for further knowledge.

With generalized peritonitis, emergency surgery is often performed to identify and treat acute pathogenesis, and to minimize bacterial load through abundant irrigation and drainage of the peritoneum [33]. A patient with generalized peritonitis often requires intensive supervision with appropriate antibiotic therapy. Generalized peritonitis can lead to septic rhinitis, which entails many adverse changes in the patient’s body. According to our study, blood cells undergo substantial ultrastructural changes, causing gas exchange disruption, changes in reactivity and decompensation of organs and systems functioning with first- or second-stage abdominal septic shock.

It has been shown that the appearance of different forms of erythrocytes is linked to reduced deformity of red blood cells, which means that their adaptability is limited and caused, for example, by polycythemia [34] or metabolic syndrome [35]. Although it is known that echinocytes may regain their discocyte forms in normal conditions, the changes are irreversible in the above-mentioned echinocytes [36]. Such forms of erythrocytes demonstrate limited abilities to alter, thus affecting intercellular contacts, triggering intravascular aggregation, which contributes to sludging, gas exchange disorders, and hypoxia. Erythrocytes do not contain the nuclei and mitochondria necessary for apoptosis, which is the programmed death of cells. However, the erythrocytes undergo a number of changes both in normal and pathological conditions distinct from apoptosis. In view of those specific mechanisms for erythrocyte apoptosis, the term ‘eryptosis’ has been introduced to describe the programmed or suicidal death of erythrocytes [37,38].

Our findings serve to demonstrate the destructive (necrotic rather than apoptotic) changes in erythrocytes caused by toxemia with distinctive disorders in both extracellular and intracellular homeostasis [38]. The ultrastructural changes in granulocytes testify to their hyperactivation. According to the authors [29], this neutrophilic state leads to the exhaustion of membrane synthetic resources, membrane destruction and ineffective expenditure of bactericidal factors on substrates not subject to destruction. Although eosinophils undergo analogous changes, they are less distinct, compared to neutrophils. 

The destructive changes to intracellular membranes in lymphocytes of children with generalized purulent peritonitis have also been described by a number of authors under other pathologies, and may be regarded as the result of profound changes in cell metabolism, the disturbance of reparative processes, and especially of energy-supplying and protein-synthesizing systems of the cell.

The strengths and weaknesses of this study merit some consideration. First of all, to our knowledge, this is the first study to investigate and present the ultrastructure of blood cells in children suffering abdominal septic shock against the background of generalized purulent peritonitis of appendicular origin. Secondly, the research results we present may have implications in everyday clinical practice when treating septic shock patients with generalized peritoneal flooding caused by acute appendicitis. Early recognition of changes at the cellular and molecular levels based on the changes in blood cells at the stage of development of symptoms of shock can constitute an essential biomarker of septicemia and its effects as well as allows for earlier implementation of therapeutic activities in the patients. Identification of accurate biomarkers that could facilitate a better choice of patients and more appropriate direction of previously ineffective therapeutic approaches can result in better results. The strength of our study is also the perspective project.

A weakness of this study was first and foremost the small size of the group of patients. Furthermore, our study did not include the control group comprising children with acute appendicitis without symptoms of generalized peritonitis. Comparing structural changes of blood cells for the two groups would serve as a stronger point and research evidence. Due to the costs of the ultrastructural imaging method used, there is a modest number of patients. The lack of the control cohort needed was dictated by the difficulty in achieving this in an emergency care setting, which has been noted by Schultz et al. in their research [39].

## 5. Conclusions

The results of this study suggest that in conditions of abdominal sepsis with signs of first- or second-stage shock, blood cells undergo substantial ultrastructural changes causing gas exchange disruption, changes in reactivity, and decompensation of organs and systems function. Changes in the ultrastructure of erythrocytes do not correlate with changes characteristic of eryptosis, thus confirming their occurrence being due to such pathogenic factors as intoxication, metabolic, water–electrolyte balance, and acid–alkaline disorders. The ultrastructural changes in granulocytes testify to their hyperactivation leading to the exhaustion of membrane synthetic resources, the destruction of membranes, and the ineffective expenditure of bactericidal factors. The findings have demonstrated disorganization of nuclear envelopes and intracellular membranes, as well as uneven distribution of chromatin, the hypertrophied Golgi apparatus, a large number of young mitochondria, lysosomes, ribosomes, and vesicles manifesting the stress and decompensation of energy-supplying and protein-synthesis systems in lymphocytes.

## Figures and Tables

**Figure 1 children-07-00189-f001:**
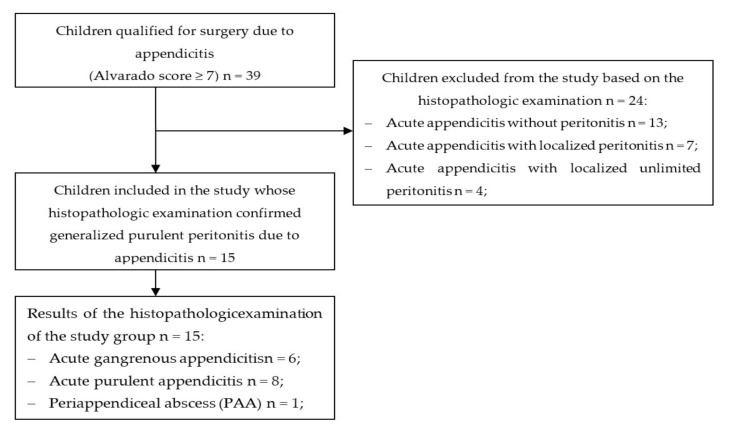
A flow-chart demonstrating the selection of the groups studied.

**Figure 2 children-07-00189-f002:**
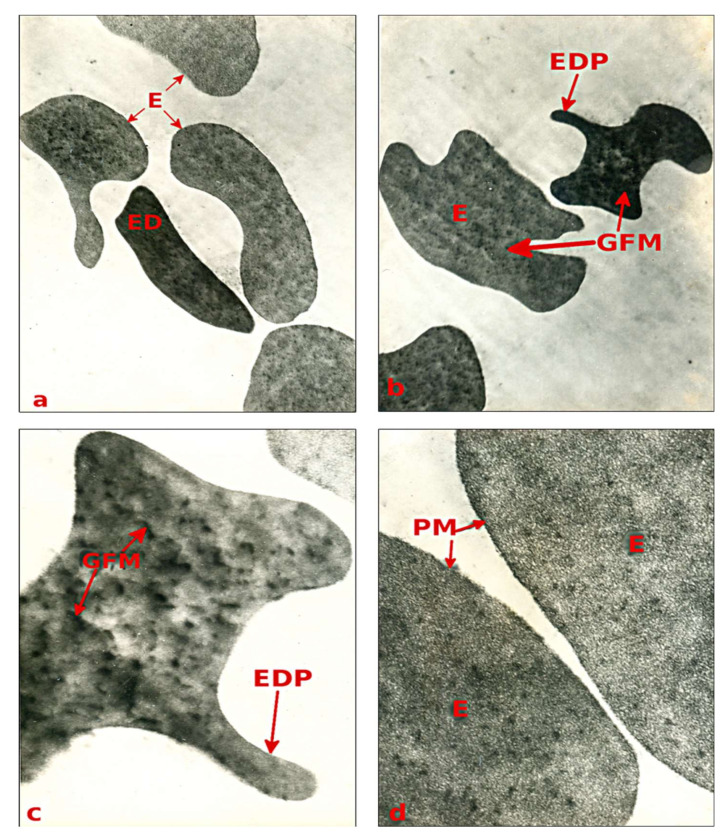
The ultrastructure of peripheral blood erythrocytes in child X, aged 12, with GPP. (**a**,**b**) erythrocytes (E) of different shapes and unequal electron density (ED), magnification of 2500×; (**c**) an electron-dense erythrocyte with digitiform projections (EDP) and available granulative-fibrillary material (GFM), magnification of 19,000×; (**d**) disorganized plasmalemma (PM) in erythrocytes of mean electron density, magnification of 23,000×.

**Figure 3 children-07-00189-f003:**
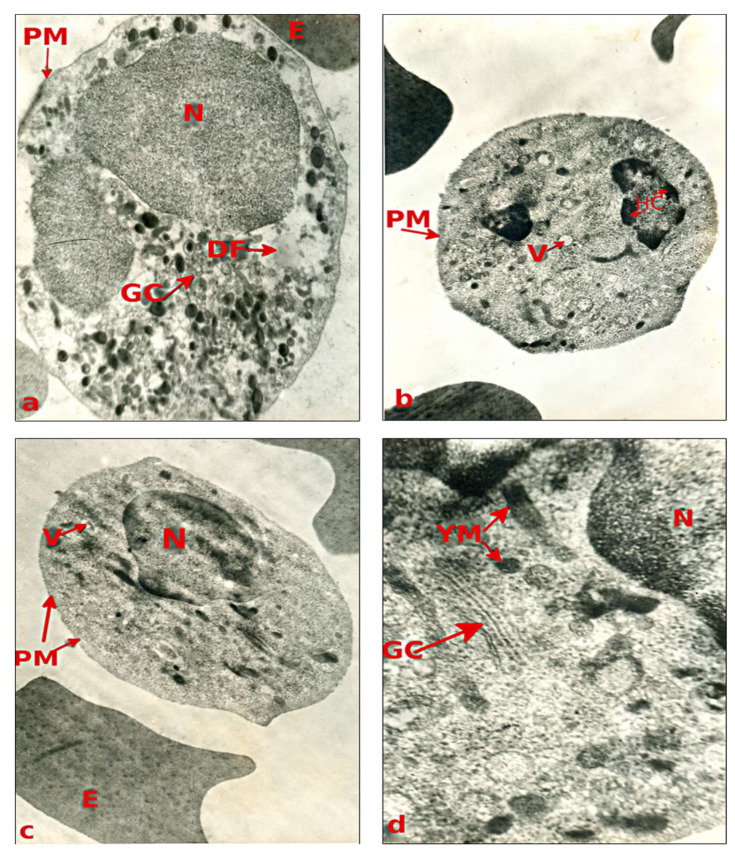
The ultrastructure of peripheral blood granulocytes and lymphocyte in child X, aged 12, with GPP. (**a**) disorganized nucleus (N) and cytoplasm of neutrophil granulocyte, magnification of 7000×; (**b**,**c**) a small number of electron-dense granules against the background of increased electron-dense cytoplasm of neutrophil granulocytes, magnification of 4500×; (**d**) fragment of lymphocyte with hypertrophied Golgo complex, magnification of 25,000×; V: vesicle; HC: heterochromatin; GC: Golgi complex; DF: drop of fat; YM: young mitochondria. The remaining abbreviations are the same as noted above.

**Table 1 children-07-00189-t001:** Basic data of children with generalized purulent peritonitis.

Variables	Study Group (n = 15)
Demographic data:	
Age [years] ^b^	9.07 ± 1.94
Gender (male) ^a^	9 (60)
Disease duration [hours] ^b^	50.67 ± 18.6
Blood count result:	
RBC [×10^12^/L] ^b^	3.69 ± 0.29
HGB [g/dL] ^b^	12.73 ± 2.71
HCT [%] ^b^	40 ± 0.5
WBC [×10^9^/L] ^b^	14.7 ± 2.86
NLR ^c^	6.1, (4.7–8.85)
PLT [×10^9^/L] ^c^	129, (90.5–140)
ESR [mm/h] ^b^	22.33 ± 5.37
Biochemical parameters:	
Total bilirubin [mg/dL] ^c^	1.7, (1.4–3.55)
CRP [mg/L] ^b^	25.23 ± 11.34

Note: Data presented as: ^a^ n(%), ^b^ mean ± SD, or ^c^ median, (Q1–Q3); RBC: red blood cells; HGB: hemoglobin; HCT: hematocrit; WBC: white blood cells; NLR: neutrophil to lymphocyte ratio; PLT: platelets; ESR: erythrocyte sedimentation rate; CRP: C-reactive protein.

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
