# Peer review of "Ultrastructural Changes of Blood Cells in Children with Generalized Purulent Peritonitis: A Cross-Sectional and Prospective Study"

_children, 2020, doi:10.3390/children7100189_

Round 1

Reviewer 1 Report

To the authors

Brief description of the study and its aim:

This was a prospective study of 15 children with complicated appendicitis. The aim of the study was to investigate the ultrastructure changes in the blood cells. No control group was included.

Revisions

  1. Consider following STROBE statement for observational studies throughout the manuscript to increase the transparency of the reporting.
  2. Title/abstract: according to STROBE mention the study design in either title or abstract
  3. Introduction: Please define abbreviation: line 55: “USG, CT, or MRI”
  4. Introduction: The introduction is rather long, but present hard any description of ultrastructure of inflammatory cells. Instead focus in on the diagnostic methods and complications of complicated appendicitis. Please consider restructuring the introduction section, so diagnosis and complications are described briefly and focus in on the aim of the study e.g. ultrastructure of blood cell and why this is interesting (from the adult literature).
  5. Methods: Please add “prospective” to the study design line 84.
  6. Was this clinical trial registered prior to inclusion e.g. at clinical trials.gov or is there a protocol available?
  7. Methods, Blood samples: When was the blood samples collected: preoperatively, perioperatively, or postoperatively?
  8. Methods, Blood samples: Which test tubes was used e.g. heparin-lithium plasma tube?
  9. Methods, Blood samples: Was the blood samples stored (temperature, light etc.) or did the undergoing analysis immediately?
  10. Methods, Other data: When was the answers of the questions collected preoperatively, perioperatively, or postoperatively?
  11. Methods: Please add a section on the statistical analysis (STROBE).
  12. Methods: Was there any follow-up of the participants?
  13. Results: You included the participants for five months, I assume not all eligible patients were included. Please describe how many patients that was eligible for the study and what was the reason for exclusion e.g. in a flow chart. Exclusions both prior to inclusion and after inclusion (as your inclusion criteria was “histopathological confirmation of acute appendicitis and generalized peritonitis” which was probably determined postoperatively) should be presented.
  14. Table 1: Your study group comprise of 15 patients. You present bot mean ± SD and median (IQR). Please only present the summary measurements relevant for your type of data (e.g. normally distributed vs. not normally distributed).
  15. Results: You write “The 127 study results revealed that all the examined children with generalized purulent peritonitis possessed 128 morphological changes of varying severity in erythrocytes, granulocytes and lymphocytes.” Change in comparison with what as you have no control population? If with definitions in the literature, these should be defined in the method section.
  16. Discussion: Please start the discussion section with a summary of the key results of the study (STROBE).
  17. Discussion: You describe “Secondly, the research results we are presenting may have implications in 212 everyday clinical practice when treating septic shock patients with generalized peritoneal flooding 213 caused by acute appendicitis.” As a strength of your study. Could you please elaborate more on why there may be an implication in everyday clinical practice based on a study of 15 children? You write that the analysis is quite expensive.
  18. Discussion: What would randomization have contributed to in your current setup – how would you have designed it otherwise?
  19. A possible strength of the study, is its prospective design.

Reviewer 2 Report

Halyuk and colleagues describe in a small study the ultrastructural changes in peripheral blood cells with focus on erythrocytes, granulocytes and lymphocytes in children with purulent peritonitis due to acute appendicitis in a pediatric patient population. The authors describe, that they observed various structural anatomic changes in these peripheral cells indicating changes to cell function in concern to gas exchange, reactivity and decompensation of organ systems. They describe that the samples from the 15 children collected are during first or second “stage” shock and sepsis.

REVIEWER COMMENTS

INTRODUCTION

  1. The approach to evaluate structural changes of peripheral cells, especially immune cells is an interesting and important approach in gain better understanding regarding the dysregulated bodies’ response to an acute infectious agent as described in the sepsis-2[1] and sepsis-3 [2,3] To describe the findings of the studies and apply the appropriate sepsis definition is important to give the article structure and guide the reader. This should be added to the description of acute appendicitis and peritonitis in the introduction.
  2. Specifically line 69 through 72 should be reviewed. SIRS (systemic inflammatory response syndrome) does not cause multiple end-organ dysfunctions and is part of a syndrome initially described by Roger Bone indicating a response and may be a first indication of the development of multi-organ dysfunction including shock [4,5].
  3. Shock is generally understood as a condition for which there is a mismatch between oxygen delivery and oxygen demand which can have various etiologies and can be signified by hypotension but also metabolic alterations such as increases in lactic acid/ lactate levels. It is a well-known observation that with hypoxemia structural changes in peripheral blood cells occur. Brief 1-2 sentences should be added to the introduction about this.[6-8]

MATERIALS and METHODS

The reviewer congratulates the research team in successfully enrolling 15 pediatric patients within less than 6 month into a study by obtaining consent from the guardians. This is surely not easy.

BLOOD SAMPLE

  1. You state that “Blood samples were collected from the posterior vein”. It is not clear what this means. Does this refer to an intraoperative blood sample from the appendicular vein? Or is this a peripheral sample from the dorsum of the hand? A clarification will be helpful.
  2. Also please describe which blood collection tubes you used (EDTA blood, Heparin Blood, Glucose blood) along with the amount of blood collected for the two tubes.

OTHER DATA

  1. Please provide the questionnaire questions and sample collection as supplemental data for review.

RESULTS

  1. Important information is missing from the table. You highlight the diagnostic work-up in the introduction nicely, however you do not provide information for CRP levels or imaging studies or the incidence of surgical intervention. If all children underwent surgical intervention just state this.
  2. CLINICAL Description needed                                                                      You also claim that the children had septic shock due to an abdominal source (ruptured appendicitis with purulent peritonitis). Data supporting the claim that this is septic shock by vital signs or laboratory assessments have not been provided and are important to understand the illness severity in these children. Data concerning the vital signs such as Heart Rate, Respiratory Rate and Temperature as part of the SIRS criteria along with information regarding Systolic Blood pressure, Diastolic Blood Pressure and Mean arterial Pressure should be provided. You may consider calculating a shock index. Lactate/ Lactic acid levels should be provided if available. It should be mentioned if vasopressors were used and how much fluids by kg bodyweight the children received. Information regarding cultures collected and culture positive results should be provided.

As the focus of the paper is the ultrastructure these information could be provided as supplement, but they are necessary to understand the severity of illness in these children and to proof that they were in septic shock.

  1. I would find a table summarizing the structural changes you describe useful, but this is not a must.
  2. Bone, R.C.; Sibbald, W.J.; Sprung, C.L. The ACCP-SCCM consensus conference on sepsis and organ failure. Chest 1992, 101, 1481-1483, doi:10.1378/chest.101.6.1481.
  3. Singer, M.; Deutschman, C.S.; Seymour, C.W.; Shankar-Hari, M.; Annane, D.; Bauer, M.; Bellomo, R.; Bernard, G.R.; Chiche, J.D.; Coopersmith, C.M., et al. The Third International Consensus Definitions for Sepsis and Septic Shock (Sepsis-3). Jama 2016, 315, 801-810, doi:10.1001/jama.2016.0287.
  4. Seymour, C.W.; Liu, V.X.; Iwashyna, T.J.; Brunkhorst, F.M.; Rea, T.D.; Scherag, A.; Rubenfeld, G.; Kahn, J.M.; Shankar-Hari, M.; Singer, M., et al. Assessment of Clinical Criteria for Sepsis: For the Third International Consensus Definitions for Sepsis and Septic Shock (Sepsis-3). Jama 2016, 315, 762-774, doi:10.1001/jama.2016.0288.
  5. Bone, R.C.; Sprung, C.L.; Sibbald, W.J. Definitions for sepsis and organ failure. Crit Care Med 1992, 20, 724-726, doi:10.1097/00003246-199206000-00002.
  6. Bone, R.C. Toward an epidemiology and natural history of SIRS (systemic inflammatory response syndrome). Jama 1992, 268, 3452-3455.
  7. Bateman, R.M.; Sharpe, M.D.; Singer, M.; Ellis, C.G. The Effect of Sepsis on the Erythrocyte. Int J Mol Sci 2017, 18, doi:10.3390/ijms18091932.
  8. Merz, T.M.; Pichler Hefti, J.; Hefti, U.; Huber, A.; Jakob, S.M.; Takala, J.; Djafarzadeh, S. Changes in mitochondrial enzymatic activities of monocytes during prolonged hypobaric hypoxia and influence of antioxidants: A randomized controlled study. Redox Rep 2015, 20, 234-240, doi:10.1179/1351000215Y.0000000007.
  9. Hotchkiss, R.S.; Moldawer, L.L.; Opal, S.M.; Reinhart, K.; Turnbull, I.R.; Vincent, J.L. Sepsis and septic shock. Nat Rev Dis Primers 2016, 2, 16045, doi:10.1038/nrdp.2016.45.

Round 2

Reviewer 1 Report

Dear authors

Your manuscript and the reporting of you methods and findings have significantly improved in the revised version.

The only comment that remains is regarding 18. Discussion: What would randomization have contributed to in your current setup – how would you have designed it otherwise? Here, you correctly answer that "Our study did not include the control group of children acute appendicitis with generalized peritonitis. Comparing structural changes of blood cells for the two groups would serve as a stronger point and research evidence." Thus my point is that a randomized design could not have improved your study a control group would. Randomization in you type of study is not possible, thus, I would advise you to change your argument in the discussion section from randomization needed to a control cohort needed.

Best of luck with the current manuscript and you future research.

Author Response

October 5, 2020

October 5, 2020

Response to the Reviewer's attention

Dear Reviewer,

We would like to thank the Reviewer for the comment with which we fully agree. To clarify the above issue, we can only say that in the multitude of changes and additions in the previous version of the revision of the article, we missed the correction of this paragraph in the discussion section, towards which we presented our position in feedback to reviewer point 18.Discussion.  We apologize for our oversight.

Our correction is marked in red – lines 277-283

Yours sincerely
